# Antimicrobial Resistance, Pharmacists, and Appreciative Inquiry: Development of a Novel Measurement Tool

**DOI:** 10.3390/antibiotics9110798

**Published:** 2020-11-11

**Authors:** Rojjares Netthong, Keivan Ahmadi, Ros Kane

**Affiliations:** 1Joseph Banks Laboratories, School of Pharmacy, University of Lincoln, Beevor St, Lincoln LN6 7DL, UK; 2Lincoln Medical School, University of Lincoln, Brayford Pool, Lincoln LN6 7TS, UK; kahmadi@lincoln.ac.uk; 3School of Health and Social Care, University of Lincoln, Brayford Pool, Lincoln LN6 7TS, UK; rkane@lincoln.ac.uk

**Keywords:** antibiotic smart use program, antibiotic stewardship, antibiotic use, antimicrobial resistance, Appreciative Inquiry, reliability, validity, community pharmacist

## Abstract

Antimicrobial resistance (AMR) is a threat to achieving the United Nation’s (UN) sustainable development goals (SDGs). The behavior of stakeholders has directly influenced the extent of AMR and understanding underpinning knowledge and attitudes is an important step towards understanding these behaviors. The aim of this study was to develop and validate a novel questionnaire, utilizing the theory of Appreciative Inquiry, to measure knowledge and attitudes around antibiotic resistance amongst community pharmacists throughout Thailand. A survey tool was developed using the Appreciative Inquiry theory, and was piloted in a non-probability sample of practicing community pharmacists. Descriptive and inferential statistics were applied and the tool validated, using a three-step psychometric validation process. A total of 373 community pharmacists participated in the study. The survey tool was found to be valid and reliable. The “Knowledge” domain of the survey tool showed an acceptable level of reliability (Cronbach’s alpha 0.64); while the “Attitude” domain showed an excellent reliability level (Cronbach’s alpha 0.84). This new survey tool has been designed to measure attitudes and knowledge of antibiotic resistance by utilizing the Discovery phase of Appreciative Inquiry theory amongst community pharmacists in Thailand. This survey tool has the potential to be used by other researchers across different settings.

## 1. Introduction

Antimicrobial resistance (AMR) is a threat to achieving the United Nation’s (UN) sustainable development goals (SDGs) [1]. It can negatively affect health security, economic growth, agriculture, and food security [1,2,3], and has already been recognized as an imminent threat to healthcare systems globally [4,5,6,7,8,9]. Evidence shows that South East Asia, with 11 low-and-middle-income countries (LMICs) is one of the highest risk regions for AMR [10]. The main concerns are the inadequacy of surveillance systems and non-prescription of antibiotic use [11]. Antibiotic prescription/dispensing in Thailand, is an interesting, yet complex issue that is highly relevant in the context of antibiotic resistance. Challenges to combatting antimicrobial resistance are inadequate surveillance systems, lack of antimicrobial consumption data, a lack of governance support related to policy research, and incomprehensive antibiotic stewardship [12]. Antibiotic prescribing and dispensing (without a doctor’s prescription) by community pharmacists are legal and common practices in Thailand [13]. Easy access to antibiotics, alongside their excessive and irrational use, have been identified and acknowledged as major contributors to the risk of developing AMR [13,14]. Both prescriber and patient/client behavior have directly influenced the extent of AMR in Thailand [15,16].

In 2016, the Thai government developed and integrated an Antibiotics Smart Use (ASU) program into its national strategic plan on antimicrobial resistance (NSP-AMR) to respond to the global action plan on antimicrobial resistance (GAP-AMR) in primary care settings [12,16]. The principal aim of ASU is to change healthcare providers’ behavior towards appropriately using the antibiotics. ASU was first implemented in secondary care settings in 2007, which has been successful in promoting rational antibiotic use through using educational interventions and financial rewards as a pay-for-performance policy to address and correct prescribers’ behavior [16].

Around 13,906 community pharmacies are private and provide a convenient first point of contact between patients and the health care system [17]. Community pharmacists play a vital role in providing healthcare services for people in community due to the convenience of their location and short waiting times [18]. Arguably, community pharmacists in Thailand are an appropriate and suitable target for a successful implementation of ASU; this is because of the current antibiotic prescribing and dispensing practices by the Thai community pharmacists. It is well established that clinical practices are informed and influenced by knowledge and attitudes [19,20,21,22,23]. Thus, it is crucial to evaluate levels of knowledge of antibiotic resistance among community pharmacists, to further explore the main reasons behind their antibiotic prescribing/dispensing attitudes and behavior [16].

Appreciate Inquiry (AI) is a theory that focuses on positive aspects of practices in work settings, including those in healthcare [24]. AI values people in organizations and draws them into innovative ideas, especially to solve issues of complex systems [25]. There is a dearth of published literature pertaining to AI in the context of community pharmacy and antibiotic practice among healthcare professionals [24,26,27,28]. This study was, therefore, conducted to examine the implementation of ASU amongst community pharmacists by applying AI theory to better understand the complexities of their prescribing/dispensing behavior.

AI consists of four main sequential and inter-related phases starting with the Discovery phase followed by the Dream phase, leading to the Design phase and eventually ending in the Destiny phase [29]. Discovery focuses on asking individuals to recall actual experiences in the best or the most effective/successful practices; and also, to focus on their aspirations to improve practices in their work settings [30]. The Dream phase allows the individual to envision ideal practice, which is heavily influenced by the individuals’ own aspirations [31]. The Design and Destiny phases focus on planning, implementing, and sustaining the desired changes, and to make the dream a reality [30].

Although other studies have developed tools to measure self-medication with antibiotics among the general public, we identified a lack of research tools to quantitatively measure community pharmacists’ attitudes and knowledge to aid the understanding of the association between the attitudes (Discovery phase) and knowledge of antibiotic resistance [32]. Such a tool has therefore been developed as a component of this study and the aim of this paper is to systematically present the processes involved in its design, development, validation and piloting for use amongst community pharmacists in Thailand. The measurement tool consists of two main phases: (a) knowledge of antibiotic resistance; and (b) attitudes towards antibiotic dispensing/prescribing based on the principles of the Discovery phase of AI theory.

## 2. Results

### 2.1. Survey Tool Design and Development

The first draft of the survey tool contained four sections:Section 1 contained 27 questions that asked about personal background, education, workplace location, and resource including antibiotic learning experience.Section 2 (the “Knowledge” domain) contained 15 statements, with three answer options to measure participants’ knowledge of the different aspects of antibiotic resistance, prescription and dispensing, epidemiology, and data relating to antibiotic use in Thailand. This section was developed from previously validated surveys used in Thailand and elsewhere [19,20,22,23,33,34,35,36,37].Section 3 (the “Attitude” domain) contained 17 questions with five-point Likert scale answer options for assessing participants’ attitudes that reflected the Discovery phase of AI theory.Section 4 contained 18 questions with five-point Likert scale answer options to assess participants’ vision of future antibiotic prescribing/dispensing in Thailand, which is reflected in the Dream phase of AI theory.

### 2.2. Translation and Back-Translation

The first version of the survey tool was translated from English into Thai through two bilingual pharmacists who were native Thai. Thai equivalents for some specialty words in English (such as secondary infection, normal flora, superbugs, DNA, and ASU) were added to retain the original meaning. After content validation by the expert committee, the second revised survey tool was back-translated by the bilingual pharmacists which resulted in an acceptable match to the original English version. The back-translation resulted in an acceptable match to the original English version.

### 2.3. Psychometric Validation

#### 2.3.1. Face Validity

The 77-item draft of the survey tool was initially evaluated for face validity. The 18 questions in Section 4 were excluded because the closed-ended questions were too restrictive and did not allow for sufficient openness or creativity in suggesting future aspirational practices to reflect the Dream phase of AI theory. Face validity assessment led to a 50-item survey tool. Some sentences were re-phrased and the order of items rearranged to improve readability and flow.

#### 2.3.2. Content Validity

Content validation was done by applying the index of item objective congruence (IOC) on each item. IOC is a method for evaluating the item quality at the developmental stages of survey tools. It basically measures the degree to which an item measures what it is meant to measure [38]. The content experts gave each item a rating of +1 (clearly measuring), 0 (unclear), and −1 (clearly not measuring) [38]. The index value was calculated as an average by dividing the total scores of the experts’ ratings in each item with the total number of the experts [38]. The 14 items of the first revised version of the survey tool obtained the index value of 1 (Table 1 and Table 2). Five statements of the knowledge section and five questions of the Discovery phase of the AI, in which the index value was less than 1, were re-phrased to make the language easier to understand and less formal. 

#### 2.3.3. Reliability

Of the 12 items comprising the “Knowledge” domains, two items were excluded from the statistical analysis. Both items could bring to inaccurate answers based on the participants’ interpretation of the different meaning of the question. One item: “Antibiotic resistance has become a public health issue” was not to be sufficiently specific as it could have been answered from both a national or global level. The other item: “Misuse of antibiotics can lead to a loss of susceptibility of a specific pathogen to an antibiotic” was open to misinterpretation in the community pharmacy settings in Thailand where no antibiotic resistance data are available. The Cronbach’s alpha value of the “Knowledge” domains was 0.64, consisting of 10 items. The “Attitude” domains reflected the Discovery phase of AI and included 12 items with a Cronbach’s alpha of 0.84, indicating a high degree of reliability. The final survey tool contained 46 items.

### 2.4. Demographic of the Participants

A total of 373 part- and full-time practicing community pharmacists across Thailand completed the revised survey tool as part of an initial pilot between 7 February and 28 July 2020. The participants’ characteristics are presented in Table 3, Table 4, Table 5, Table 6 and Table 7. Snowball sampling was conducted through social media users sharing the link widely. The participants were located across the country, in central (*n* = 144, 38.61%), northeastern (*n* = 113, 30.29%), eastern (*n* = 37, 9.92%), southern (*n* = 37, 9.92%), and western (*n* = 5, 1.34%) regions. Almost sixty percent of participants were female aged less than 30 (*n* = 108, 28.95%), with a range between 30 and 39 years (*n* = 107, 28.69%). Almost all of the participants held a pharmacy bachelor degree from 18 universities in Thailand (*n* = 369, 98.93%), only four participants graduated from overseas universities with their major clerkship in community pharmacy and hospital pharmacy (*n* = 319, *n* = 308, respectively). Nearly fifty three percent of the participants had a BSc in Pharmacy, while a quarter had a Pharm D in Pharmaceutical Care degree. The remaining participants (11.53%) had other pharmacy degrees. More than eighty percent of the participants did not have postgraduate qualifications (*n* = 307, 82.31%). More than half had a mean of 5.65 years’ experience as a community pharmacist. More than sixty percent had a single role in their community pharmacies as owner (*n* = 29, 7.77%), manager (*n* = 4, 1.07%), or staff pharmacist (*n* = 200, 53.62%). Nearly thirty eight percent of the participants had more than one role in the community pharmacy in which they were employed. In terms of antibiotic prescription/dispensing, the main reason for the pharmacist to prescribe antibiotics without prescription was an attempt to address possible infections with signs and symptoms (*n* = 329) followed by a demand by patients/clients (*n* = 176). Lack of patient willingness to consult a physician was found to be the main reason for accessing antibiotics without prescription (*n* = 299), followed by the belief that antibiotics can speed up recovery (*n* = 185). Nearly fifty five percent of the participants had antibiotic stewardship training during their pharmacy course (*n* = 202, 54.16%), while nearly a quarter had been trained after they had qualified as pharmacist (*n* = 115, 30.83%). Participants cited the Centre of Continuing Professional Education (CCPE) and training sessions as the most common sources of obtaining/updating knowledge on antibiotics.

### 2.5. Final Survey Tool

The final survey tool was established as a new, Appreciative Inquiry theory, self-administered online survey tool. It consisted of the following three sections:Section 1, containing 24 questions that asked about personal background, education, workplace location, and resource including antibiotic learning experience.Section 2, or the “Knowledge” domain, containing 10 valid and reliable statements with three response options to measure participants’ knowledge of the different aspects of antibiotic resistance, prescription and dispensing, epidemiology, and data relating to antibiotic use in Thailand. This section was developed from previously validated surveys used in Thailand and elsewhere [19,20,22,23,33,34,35,36,37].Section 3, or the “Attitude” domain, containing 12 valid and reliable questions with five-point Likert scale answer options for assessing participants’ attitudes that reflected the Discovery phase of AI theory.

## 3. Discussion

Mobilizing stakeholders has proven to be one of the most effective approaches to tackling the current challenges with AMR and to achieve the United Nations’ SDGs [39]. To mobilize stakeholders i.e., healthcare professionals and patients/clients in the fight against AMR, it is crucial to measure their baseline levels of knowledge, attitudes, and practices to design tailored interventions for improvement [35,40]. We have successfully developed and piloted a new measurement tool by applying the principles of AI.

### 3.1. Survey Tool Design and Development

Following a careful review of literature, a lack of understanding of the influences on antibiotic prescribing and dispensing among community pharmacists in Thailand was identified. Understanding underpinning knowledge and attitudes is an important step towards understanding behaviors and this survey tool has been developed to facilitate this data collection. Previous studies have surveyed national community pharmacists about AMR, behavior theories such as knowledge, attitudes and practices (KAP), knowledge, perceptions, and practices (KPP) showed that the pharmacists’ knowledge and attitudes could relate and influence their practice [19,36]. However, they could not determine the direct relationship between the knowledge and practices of community pharmacists [23]. The reasons behind their antibiotic dispensing attitudes and behavior might have been related to the socio-demographic characteristics or might have been relevant to their dealings with the regulatory aspects of the practices [21,37]. In this study a new measurement tool has been successfully developed and piloted that could be used to reliably measure the knowledge, attitudes and practices of community pharmacists [19,20,22,23,33,34,35,36,37]. Moreover, the “Attitude” domain of this new survey tool, that allows the measurement of the Discovery phase of the AI, was found to be highly reliable in measuring attitudes regarding their appreciation and awareness of the current challenges of AMR in local practices. The strength of this novel survey tool is the positive wording of the “Attitude” items that reflect the Discovery phase of AI theory could promote participants’ desire to further involve themselves with the whole AMR problem. This is an important first step in changing conventional practices and habits towards innovative problem-solving approaches to complex issues [26].

To the best of our knowledge, AI theory has not previously been applied in community pharmacy research. This is the first research to literature the AI approach in the community pharmacy setting [24,26,27,28]. It is strength-based research that offers people in the community or organization to share their knowledge and build new opportunities in their community or their organization. Especially, community pharmacists who are professional pharmacy services in community pharmacy, must be willing about caring professionals and enhancing quality of their services.

### 3.2. Psychometric Validation

Psychometric validation was conducted using three different methods. It was piloted amongst 373 Thai community pharmacists as the target population resulting in a fully validated survey tool for assessing attitudes and knowledge about antibiotic resistance amongst the community pharmacists’ in Thailand. The knowledge section showed acceptable reliability. The five attitude scales demonstrated satisfactory internal validity as evident from the results of the Discovery phase of AI, and a good reliability demonstrated by a Cronbach’s alpha of >0.8. It means our survey tool displayed good functioning to measure attitudes and knowledge [41]. The final 46 items of the survey tool consist of 24 demographic items, 10 developed and modified in the knowledge section, and 12 innovative in the Discovery phase of AI theory.

### 3.3. Limitations of the Study

#### 3.3.1. Generalizability and Transferability of the Tool

This survey tool was developed with a focus on the current community pharmacy practices in Thailand, where community pharmacists can legally prescribe/dispense antibiotics. Prior to using this survey tool elsewhere, cross-cultural adaptation and validation should be undertaken. We would highly recommend the use of this tool to the researchers in their future work, where it should be utilized in a representative sample of community pharmacists as well as hospital pharmacists.

#### 3.3.2. Possible Measurement Bias as a Result of Applying AI Theory

It should also be noted that, AI, by definition, focuses on the strengths of the current antibiotic prescribing/dispensing practices and does not focus on the problems/issues. Hence, this tool could be less sensitive in identifying and measuring the current shortcomings of the antibiotic prescribing/dispensing practices of the community pharmacists. Nevertheless, it is important to re-emphasize the usefulness of this tool as a good starting point bring about behavior change, an affordable and a sustainable intervention, to tackle antimicrobial resistance in developing countries where the resources are scarce.

#### 3.3.3. Participant Recruitment Bias

In response to the current COVID-19 pandemic, we had to collect data via Facebook and Line, instead of using the traditional paper and pencil method of data collection. It has been shown that online data collection, especially via social media platforms, could introduce recruitment bias, as the use of such platforms have been associated with certain socio-demographic characteristics [42]. However, after analyzing the participants’ characteristics (Table 3, Table 4, Table 5, Table 6 and Table 7), it seemed that the participants represented Thai community pharmacists well [43].

## 4. Methods

### 4.1. Study Population and Survey Tool Administration

Part- and full-time practicing community pharmacists across Thailand represent the population in the study. Pharmacies are classified into two categories: type 1 community pharmacy, the predominant type that requires a registered pharmacist to be present during working hours, and type 2 pharmacy, which does not require pharmacist to be on duty [44]. This study focused on type 1 community pharmacies that had permission for pharmacists to dispense antibiotics without a physician’s prescription [45].

The survey tool was disseminated via a non-probability sampling technique using social networks, such as Facebook^TM^ and Line^TM^ [43]. The survey link was open to anyone who could access these social networks and users were invited to share it [46]. The link contained information about the target audience and the specific eligibility criteria for potential participants to ensure that social media users, who were not practicing Thai community pharmacists, did not participate. Furthermore, the first two items in the tool helped to screen out those who did not meet the inclusion criteria (see Appendix A).

### 4.2. Survey Tool Design and Development

#### 4.2.1. Hypothesis

Initially, an extensive review of published academic literature to determine the need to develop a new survey tool was conducted, and the relevant literature was used to inform the content of the survey tool. Studies were screened by determining the factors associated with antibiotic prescribing/dispensing amongst healthcare professionals [19,20,22,23,33,34,35,36,37]. Following a careful review of literature, it became evident that there was a need to develop a tool to quantitatively measure the community pharmacists’ attitudes with regard to the Discovery phase of the AI [47]. Moreover, a lack of an existing tool to measure knowledge of antibiotic resistance amongst community pharmacists was noted. Therefore, we hypothesized that the Appreciate Inquiry could be used to create tools to quantitatively measure community pharmacists’ attitudes towards antibiotic smart use in Thailand. Item generation, reduction, and development of the survey tool are demonstrated in Table 1 and Table 2.

#### 4.2.2. Theoretical Framework

The principles of Appreciative Inquiry (AI) were applied, as the theoretical framework for the design of the survey tool (Figure 1). AI consists of four main sequential and inter-related phases, starting with the Discovery phase, followed by a Dream phase, leading to a Design phase, and eventually ending in a Destiny phase [30]. The Discovery phase focuses on asking individuals to recall actual experiences of best or the most effective practice; and also to focus on their aspirations to improve practice in their work settings [30]. This novel approach allowed the creation of a tool to capture data that were informative and relevant to antibiotic dispensing/prescribing knowledge, attitudes, and practices among Thai community pharmacists.

### 4.3. Psychometric Validation

The validity of the survey tool was psychometrically tested following a three-step approach (Figure 1).

#### 4.3.1. Face Validity

Face validity aimed to examine the suitability of the survey tool’s structure, appearance, feasibility, readability, consistency of style between questions, formatting, and clarity of language [47,48,49]. The first draft of the survey tool had four sections, containing 77 items, which were validated by three academic pharmacists. Their input informed revision of items to improve the potential for cross-culture adaptation, appropriateness of the response options, and also ease of completion.

#### 4.3.2. Content Validity

Content validity is useful to assess the measurement capabilities of each of the items of the “Knowledge” and the “Attitudes” domain of the survey tool [48]. Content validity was assessed by three experts in community pharmacy and pharmacy education in Thailand. This was done by applying the index of item objective congruence (IOC) on each item. IOC is a method by evaluating individual items based on the degree to measure the objectives of the using survey tool [38]. The content experts gave each item a rating of +1 (clearly measuring), 0 (unclear), and −1 (clearly not measuring) [38]. The index value was calculated as an average by dividing the total scores of the experts’ ratings in each item with the total number of the experts [38]. An IOC value of 1 indicated the panel of content experts all rated the content validity as “very good to measure”. The cut off IOC score was 0.5 [38]. However, all items in which the index value was less than 1 were revised to ensure that the survey tool items were legible and capable of measuring the knowledge, attitudes, and practices of antibiotic dispensing/prescribing of community pharmacists in Thailand.

#### 4.3.3. Reliability Testing

Cronbach’s alpha is widely used to test reliability for evaluating the homogeneity of the question items in each domain [41]. The Cronbach’s alpha is a measure to establish internal consistency and reproducibility; and ranges between 0 and 1 [41,50]. In an ideal world, a measurement tool with score of 1 is considered to have reached the best possible reliability value. However, in reality, the acceptable values of the Cronbach’s alpha would depend on a myriad of factors, relevance of the measurement tool to a particular research question being one of the most relevant ones [41]. It is widely agreed that an acceptable Cronbach’s alpha score should be between 0.60 and 0.8. A Cronbach’s alpha value of less than 0.5 is usually unacceptable [51,52].

### 4.4. Pilot Test

A pilot survey was carried out for testing the correctness of the instructions and can also help to explore possible issue impacts to effective survey results prior to the large-scale. It was used to measure how well the participants in the pilot sample followed survey tool’s directions [53]. The survey tool was piloted on a sample of 373 Thai community pharmacists to clarify each item [54]. The pilot test was conducted from 7 February to 28 July 2020.

### 4.5. Translation and Back-Translation

The survey tool was initially developed in the English language, and then translated into Thai by the lead researcher (RN) and a Thai native bilingual academic pharmacist. After the content validation, the survey tool was then back-translated into English by a bilingual (Thai and English) pharmacist who had not previously seen the English version. RN and the supervisory team compared the back-translated survey tool, with the initial English version of the survey tool for conceptual equivalence and cross-cultural understanding to ensure retention of the correct meaning, and cultural relevance [50].

### 4.6. Data Analysis

Descriptive and inferential statistics including reliability analyses were performed using RStudio and “psych” package, an open access statistical computing software package [55,56]. Before analysis, data validating and cleaning was performed to remove inaccurate records from a database as csv file and then coding data was prepared. Three scales “yes-no-don’t know” answer options of the knowledge domain were determined by giving one score for correct answer and zero score for incorrect and “don’t know” answer. Five-point Likert scale “Very good-Poor” answer options of the attitude domain were re-coded by giving four for a “very good” answer and zero for “very poor”, respectively. Then, the knowledge and attitude scores were calculated to 100.

### 4.7. Ethics

The research was granted approvals from the University of Lincoln Human Ethics Committee (Ethics Reference: 2019-Jul-0366) and the Ubon Ratchathani University Research Ethics Committee (Ethics Reference: UBU-REC—33/2562).

## 5. Conclusions

A valid and reliable novel survey tool has been tested to measure Thai community pharmacists’ knowledge, attitudes, and practices regarding antibiotic smart use. Parts of this novel measurement tool were developed by applying the Discovery phase of AI theory for the first time. Other researchers are encouraged to also test the validity and reliability of this measurement tool to further improve its psychometric properties and applicability in other settings.

## Figures and Tables

**Figure 1 antibiotics-09-00798-f001:**
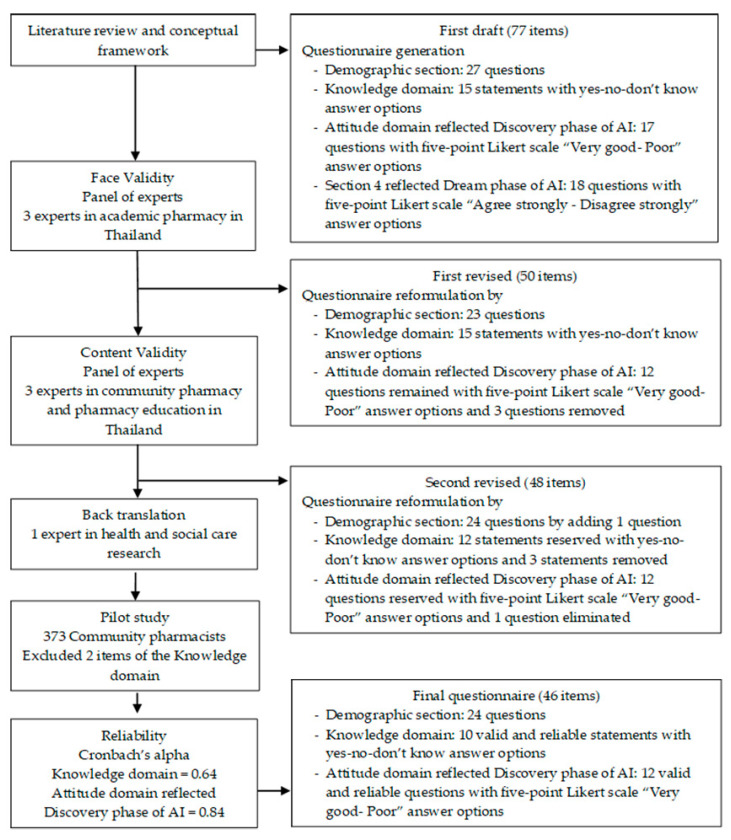
Flow Chart of the Development and Validation of the Novel Survey Tool.

**Table 1 antibiotics-09-00798-t001:** Lists of Items of Knowledge Domain of the Survey Tool.

Question/Statement	Psychometric Validation		
	Face Validation	Content Validation (IOC Index *^1^*)	Reliability Testing
Knowledge Section			0.64
Dispensing antibiotics without a prescription is a legal practice in Thailand.	Accept	1	
Dispensing antibiotics without a prescription is a common practice among community pharmacists in Thailand.	Accept	0.67	
Dispensing antibiotics without a prescription is contributing to the development of antibiotic resistance.	Accept	0.67	
Dispensing antibiotics without a prescription is contributing to the inappropriate use of antibiotics by patients.	Accept	0.67	
Antibiotic resistance has become a public health issue.	Accept	1	Exclude
Antibiotics are indicated to reduce any kind of pain and inflammation.	Accept	0.67	
Antibiotics are useful for bacterial infections.	Accept	0.67	
Antibiotics are useful for viral infections.	Reject		
Antibiotics can cause secondary infections after killing the normal flora of the human body.	Accept	1	
Antibiotics can cause allergic reactions.	Reject	-	
One should stop taking antibiotics before the completion of a full course of antibiotic therapy, if symptoms improve.	Reject	-	
Misuse of antibiotics can lead to a loss of susceptibility of a specific pathogen to an antibiotic.	Accept	1	Exclude
Superbugs are microorganisms which generate antimicrobial resistance. They include bacteria, fungal, viruses or parasites.	Accept	1	
Resistance DNA in bacteria can transfer to other bacteria by a virus.	Accept	1	
The main objective of antibiotic stewardship is the achievement of the most effective clinical outcome with the least adverse reactions.	Accept	1	

***^1^*** Item Objective Congruence (IOC) index is a method for evaluating the item quality at the developmental stages of survey tools.

**Table 2 antibiotics-09-00798-t002:** Lists of Items of Attitude Domain of the Survey Tool.

Question/Statement	Psychometric Validation		
	Face Validation	Content Validation (IOC Index *^1^*)	Reliability Testing
Discovery			0.84
How do you rate the implementation of local guidelines, before dispensing antibiotics?	Accept	1	
How do you rate the implementation of local guidelines, before prescribing antibiotics?	Reject	-	
How do you rate the clarity of the advice given to the patients about their dispensed antibiotics?	Accept	1	
How do you rate the completeness of the advice given to the patients about their dispensed antibiotics?	Accept	0.67	
How do you rate the acknowledgment of the patients’ understanding of the advice given to them about their dispensed antibiotics?	Accept	1	
How do you rate the answering of patients’ questions about their dispensed antibiotics?	Accept	0.67	
How do you rate the manager’s understanding of antibiotic stewardship?	Reject	-	
How do you rate the owner’s understanding of antibiotic stewardship?	Reject	-	
How do you rate patients’ satisfaction with antibiotic dispensing?	Accept	0.67	
How do you rate patients’ satisfaction with your antibiotic prescribing practices?	Reject	-	
How do you rate the patients’ knowledge about antibiotic stewardship?	Accept	0.67	
How do you rate antibiotic inventory management?	Reject	-	
How do you rate the support to implement antibiotic stewardship?	Accept	1	
How do you rate engagement with antibiotic awareness campaigns?	Accept	1	
How do you rate engagement with health promotion campaigns on prevention/reduction transmission of infection?	Accept	1	
How do you rate collaboration with other healthcare professionals to implement antibiotic stewardship?	Accept	1	
How do you rate the relationship between clients/patients and pharmacists in regards with antibiotic stewardship?	Accept	0.67	

***^1^*** Item Objective Congruence (IOC) index is a method for evaluating the item quality at the developmental stages of survey tools.

**Table 3 antibiotics-09-00798-t003:** Demographic of the Participants.

Characteristics	*n* = 373 (%)
Gender and Age	
Male	128 (34.32)
Less than 30	48 (12.87)
30–39	49 (13.14)
40–49	16 (4.29)
50–59	10 (2.68)
60 and above	5 (1.34)
Female	245 (65.68)
Less than 30	108 (28.95)
30–39	107 (28.69)
40–49	25 (6.70)
50–59	3 (0.80)
60 and above	2 (0.54)
Postgraduate qualification	
Yes	66 (17.69)
No	307 (82.31)
Location in Thailand	
Northern	21 (5.63%)
Northeastern (Isan)	113 (30.29%)
Western	5 (1.34%)
Central	144 (38.61%)
Eastern	37 (9.92%)
Southern	37 (9.92%)

**Table 4 antibiotics-09-00798-t004:** Professional Background of the Participants.

Characteristics	*n =* 373 (%)
Pharmacy University	
Burapha University	5 (1.34)
Chiang Mai University	28 (7.51)
Chulalongkorn University	25 (6.70)
Eastern Asia University	4 (1.07)
Huachiew Chalermprakiet University	18 (4.83)
Khon Kaen University	28 (7.51)
Mahasarakham University	17 (4.56)
Mahidol University	20 (5.36)
Naresuan University	20 (5.31)
Payap University	5 (1.34)
Prince of Songkla University	14 (3.75)
Rangsit University	31 (8.31)
Siam University	5 (1.34)
Silpakorn University	28 (7.51)
Srinakharinwirot University	10 (2.68)
Ubon Ratchathani University	97 (26.01)
University of Phayao	7 (1.88)
Walailak University	7 (1.88)
University outside Thailand	4 (1.07)
Undergraduate degree	
BSc in Pharmacy	196 (52.55)
BSc in Pharmacy and Doctor of Pharmacy: Pharm D (Pharmaceutical Care)	10 (2.86)
BSc in Pharmacy: Pharm D (Industrial Pharmacy)	3 (0.80)
BSc in Pharmacy, Doctor of Pharmacy: Pharm D (Pharmaceutical and Health Consumer Protection)	1 (0.27)
Doctor of Pharmacy: Pharm D (Pharmaceutical Care)	134 (35.92)
Doctor of Pharmacy: Pharm D (Pharmaceutical Care) and Doctor of Pharmacy: Pharm D (Industrial Pharmacy)	1 (2.68)
Doctor of Pharmacy: Pharm D (Pharmaceutical Care) and Doctor of Pharmacy: Pharm D (English Program)	1 (2.68)
Doctor of Pharmacy: Pharm D (Industrial Pharmacy)	25 (6.70)
Doctor of Pharmacy: Pharm D (Pharmaceutical and Health Consumer Protection)	1 (2.68)
Doctor of Pharmacy: Pharm D (English Program)	1 (2.68)
Clerkship ^a^	
Community pharmacy	319
Hospital pharmacy	308
Manufacturing	100
Drug registration	28
Regulation and jurisdiction	25
Research and development	53
Other	22

^a^ Multiple answers accepted.

**Table 5 antibiotics-09-00798-t005:** Employment Detail of the Participants.

Characteristics	*n* = 373 (%)
Experience as community pharmacist (years)	
Mean	5.65
Median	3
Role in community pharmacy	
Owner	29 (7.77)
Owner, manager and staff pharmacist	32 (8.58)
Owner and staff pharmacist	76 (20.38)
Manager	4 (1.07)
Manager and staff pharmacist	32 (8.58)
Staff pharmacist	200 (53.62)
Additional Role ^a^	
Hospital pharmacist	150
Pharmaceutical sales and marketing representative	28
Industrial pharmacist	17
Teacher/research in academia/university	24
Researcher at research and development	6
Registration pharmacist	15
Consumer protection officer	28
Policy maker	1
Antibiotic smart use/antimicrobial resistance committee	1
Community pharmacy (only)	130
Other	5
Type of pharmacy	
Independent store	247 (66.22)
Chain store	126 (33.78)

^a^ Multiple answers accepted.

**Table 6 antibiotics-09-00798-t006:** Reasons of using antibiotics without prescription.

Characteristics	*n* = 373 (%)
Reasons to patients/client access antibiotics without prescription ^a^	
Lack of patient willingness to consult a physician for a non-serious infection	299
Inability to afford a consultation with a physician	113
Inability to travel to other clinics	128
Keeping leftover antibiotics for future use	139
Sharing antibiotics with others	19
Belief that antibiotics can speed up recovery	185
Belief that antibiotics can eradicate any infection	124
Other	18
Reason for pharmacist to prescribe antibiotics without prescription ^a^	
Demand by the manager	16
Demand by the owner	24
Demand by patients/clients	176
Patients/clients’ satisfaction	86
Fear of losing patients/clients	33
Sales offers and discounts by pharmaceutical companies	4
Increase sales/profit-seeking	19
Indicate possible infection with signs and symptoms	329
Other	4

^a^ Multiple answers accepted.

**Table 7 antibiotics-09-00798-t007:** Antibiotic Stewardship Training Experience.

Characteristics	*n* = 373 (%)
Antibiotic stewardship training experience	
During pharmacy course	
No	94 (25.20)
Yes	202 (54.16)
Not sure	77 (20.64)
Since pharmacy degree qualified	
No	258 (69.17)
Yes	115 (30.83)
Sources of knowledge ^a^	
Training session	265
Special literature	237
Patient information leaflet	134
Sales representative from pharmaceutical company	104
Articles in CCPE (center of continuing professional education)	273
Guidelines for the diagnosis and antibiotic treatments	249
Other	9

^a^ Multiple answers accepted.

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
