# Peer review of "Antimicrobial Resistance, Pharmacists, and Appreciative Inquiry: Development of a Novel Measurement Tool"

_antibiotics, 2020, doi:10.3390/antibiotics9110798_

Round 1

Reviewer 1 Report

Dear Authors,

The manuscript ID: antibiotics-978192-peer-review-v1 entitled “Antimicrobial Resistance, Pharmacists and Appreciative Inquiry: Development of a Novel Measurement Tool” written by Keivan Ahmadi, Ros Kane and Rojjares Netthong is original.

The subject of this article is topical. Antibiotic resistance is one of the biggest public health challenges of our time. In this work, a novel survey tool has been tested to measure Thai community pharmacists’ knowledge, attitudes and practices regarding antibiotic smart use. A survey tool was developed using Appreciative inquiry theory and was piloted utilizing a non-probability sampling technique among practicing community pharmacists throughout Thailand. This manuscript is interesting.

However, according to me these data aren’t enough to be published in such a prestigious journal as “Antibiotics”.

With highest regards,

Author Response

We are thankful to the reviewers who have kindly commented on our manuscript with the editorial number antibiotics-978192.

After a thorough and careful consideration of the reviewers’ comments, we have presented our itemised responses in a tabular form for your kind perusal.

Reviewers’ comments Our response Changes in revised manuscript

Review 1

Point 1:

“The manuscript is …original. The subject of this article is topical…, a novel survey tool has been tested to measure, antibiotic smart use…”
We appreciate the reviewer’s holistic view regarding the global importance of antibiotic resistance and the immediate importance/relevance of our study that has brought about the development of a valid, reliable and innovative measurement tool, which could potentially help the researchers, clinicians and academics to further investigate the approaches/methods to tackle global antibiotic resistance. We are also glad that the originality of our work has been recognised by the reviewer. None

Reviewer 1

Point 2:

These data aren’t enough to be published in such a prestigious journal as “Antibiotics”.

We concur with the reviewer that Antibiotics is such a prestigious journal that has been helping in the dissemination of peer-reviewed high-quality information regarding the different aspects of antibiotic resistance.

However, we would like to address the concerns on the quantity of data via:

  • Gently reminding the reviewer that there have been similar publications on piloting new measurement tools that have been successfully published in the Antibiotics journal in the recent months (February 2020) that have reported their findings from 100 participants. This is the link to the above-mentioned article: https://pubmed.ncbi.nlm.nih.gov/32102325/. We have referred to this study included this study in the introduction section of the manuscript (line 79 – 80, page 2, introduction section).
  • We have added additional data points by including the information from 373 participants. This means that we have added 263 participants to the initial reported pilot from 110 participants. As a result we have updated the information under the 2.5 subheading “Demographic of the participants” in page 7 as well as the information in Tables 3 to 7 on pages 7 - 12.

Please refer to the lines 67 to 166 on pages 2- 12 under the introduction and the results sections to view the track changes.

Regarding more minor matters, we have now added some sentences and have changed order of the authors list, our spelling and phrasing patterns. The changes are highlighted using track changes option as well as being highlighted in yellow. We have provided the line number along with page number for each response, where possible, to facilitate navigation through the revised manuscript.

Keywords

We have added “community pharmacist”. (line 32, page 1)

Introduction

The introduction could be much more clear. We have included sentences such as:

“The main concerns are the inadequacy of surveillance systems and non-prescription of antibiotic use [11].” (line 39 – 40, page 1)

“Challenges to combatting antimicrobial resistance are inadequate surveillance systems, lack of antimicrobial consumption data, a lack of governance support related to policy research and incomprehensive antibiotic stewardship [12].” (line 42 – 44, page 2)

“Around 13,906 community pharmacies are private and provide a convenient first point of contact between patients and the health care system [17]. Community pharmacists play a vital role in providing healthcare services for people in community due to the convenience of their location and short waiting times [18].” (line 56 – 59, page 2)

“There is a dearth of published literature pertaining to AI in the context of community pharmacy and antibiotic practice among healthcare professionals [24,26–28].“ (line 67 – 69, page 2)

“Although other studies have developed tools to measure self-medication with antibiotics among the general public” (line 79 – 80, page 2)

Discussion

To address the strength of the novel survey tool, a new paragraph has been added to sub-heading 3.1. “Survey tool design and development “ in line 217 -222 on page 13.

“To the best of our knowledge, AI theory has not previously been applied in community pharmacy research. This is the first research to literature the AI approach in the community pharmacy setting [24,26–28]. It is strength-based research that offers people in the community or organization to share their knowledge and build new opportunities in their community or their organization. Especially, community pharmacists who are professional pharmacy services in community pharmacy, must be willing about caring professionals and enhancing quality of their services.”

We hope the revised manuscript will better suit the Antibiotics but are happy to consider further revisions, and we thank you for your continued interest in our research.

Sincerely,

Rojjares Netthong (On behalf of the Research Team)

School of Pharmacy

University of Lincoln,

Joseph Banks Laboratories

Beevor St, Lincoln LN6 7DL, United Kingdom

Mobile Tel: +44 7518626004

03 November 2020

Reviewer 2 Report

The authors presenta nicely the results of a well conducted study on the measurement of  attitudes and knowledge of antibiotic resistance by utilizing the Discovery phase of Appreciative inquiry theory amongst community pharmacists in Thailand.

Minor comments:

I would suggest to authors to briefly explain the use of Cronbach’s alpha in the methods section

The authors should comment on the limitatiosn of the study in the discussion

Author Response

We are thankful to the reviewers who have kindly commented on our manuscript with the editorial number antibiotics-978192.

After a thorough and careful consideration of the reviewers’ comments, we have presented our itemised responses in a tabular form for your kind perusal.

Reviewers’ comments Our response Changes in revised manuscript

Review 2

Point 1:

The authors present[ed] nicely the results of a well conducted study on the measurement

We really appreciate the very positive comments regarding the design of the study, its execution and lastly the presentation of the findings.

We have been very thorough from the set out of the study; and we feel reassured by the reviewer’s comments that our manuscript clearly reflects a quality work.  
None

Review 2

Point 2:

Briefly explain the use of Cronbach’s alpha in the methods section

We have expanded on this point to explain the Cronbach’s alpha in terms of use and interpretation. The new sentence read as follows: “Cronbach’s alpha is widely used to test reliability for evaluating the homogeneity of the question items in each domain [40]. The Cronbach’s alpha is a measure to establish internal consistency and reproducibility; and ranges between 0 and 1 [40,50]. In an ideal world a measurement tool with score of 1 is considered to have reached the best possible reliability value. However, in reality the acceptable values of the Cronbach’s alpha would depend on a myriad of factors, relevance of the measurement tool to particular research question being one of the most relevant one [40]. It is widely agreed that an acceptable Cronbach’s alpha score should be between 0.60 and 0.8. A Cronbach’s alpha value of less than 0.5, is usually unacceptable [51,52].”

Please refer to the lines 319 to 327 on pages 17 under the methods section to view the track changes.

Reviewer 2

Point 3

The authors should comment on the limitations of the study in the discussion

We have noted that both reviewers 2 & 3 have made a similar comment regarding the need for a more explicit declaration of the limitation(s) of our study. We have added a number of subsections with informative headings that summarise key points in the discussion.

To address the limitations, we have added the below three main plausible limitations:

3.3. Limitations of the study

3.3.1 Generalizability and transferability of the tool

This survey tool was developed with a focus on the current community pharmacy practices in Thailand, where community pharmacists can legally prescribe/ dispense antibiotics. Prior to using this survey tool elsewhere, cross-cultural adaptation and validation should be undertaken. We would highly recommend the use of this tool to the researchers in their future work, where it should be utilized in a representative sample of community pharmacists as well as hospital pharmacists.

3.3.2 Possible measurement bias as a result of applying AI theory:

It should also be noted that, AI, by definition, focuses on the strengths of the current antibiotic prescribing/dispensing practices and does not focus on the problems/issues. Hence this tool could be less sensitive in identifying and measuring the current shortcomings of the antibiotic prescribing/dispensing practices of the community pharmacists. Nevertheless, it is important to re-emphasize the usefulness of this tool as a good starting point bring about behavior change, an affordable and a sustainable intervention, to tackle antimicrobial resistance in developing countries where the resources are scarce.

3.3.3 Participant recruitment bias

In response to the current COVID-19 pandemic, we had to collect data via Facebook and Line, instead of using the traditional paper and pencil method of data collection. It has been shown that online data collection, specially via social media platforms could introduce recruitment bias as the use of such platforms have been associated with certain socio-demographic characteristics [41]. However, after analysing the participants, characteristics (See table 3 - 7) it seemed that the participants represented Thai community pharmacists well [42].

Refer to the lines 234 to 255 on pages 14 under the discussion section to view the track changes.

Regarding more minor matters, we have now added some sentences and have changed order of the authors list, our spelling and phrasing patterns. The changes are highlighted using track changes option as well as being highlighted in yellow. We have provided the line number along with page number for each response, where possible, to facilitate navigation through the revised manuscript.

Keywords

We have added “community pharmacist”. (line 32, page 1)

Introduction

The introduction could be much more clear. We have included sentences such as:

“The main concerns are the inadequacy of surveillance systems and non-prescription of antibiotic use [11].” (line 39 – 40, page 1)

“Challenges to combatting antimicrobial resistance are inadequate surveillance systems, lack of antimicrobial consumption data, a lack of governance support related to policy research and incomprehensive antibiotic stewardship [12].” (line 42 – 44, page 2)

“Around 13,906 community pharmacies are private and provide a convenient first point of contact between patients and the health care system [17]. Community pharmacists play a vital role in providing healthcare services for people in community due to the convenience of their location and short waiting times [18].” (line 56 – 59, page 2)

“There is a dearth of published literature pertaining to AI in the context of community pharmacy and antibiotic practice among healthcare professionals [24,26–28].“ (line 67 – 69, page 2)

“Although other studies have developed tools to measure self-medication with antibiotics among the general public” (line 79 – 80, page 2)

Discussion

To address the strength of the novel survey tool, a new paragraph has been added to sub-heading 3.1. “Survey tool design and development “ in line 217 -222 on page 13.

“To the best of our knowledge, AI theory has not previously been applied in community pharmacy research. This is the first research to literature the AI approach in the community pharmacy setting [24,26–28]. It is strength-based research that offers people in the community or organization to share their knowledge and build new opportunities in their community or their organization. Especially, community pharmacists who are professional pharmacy services in community pharmacy, must be willing about caring professionals and enhancing quality of their services.”

We hope the revised manuscript will better suit the Antibiotics but are happy to consider further revisions, and we thank you for your continued interest in our research.

Sincerely,

Rojjares Netthong (On behalf of the Research Team)

School of Pharmacy

University of Lincoln,

Joseph Banks Laboratories

Beevor St, Lincoln LN6 7DL, United Kingdom

Mobile Tel: +44 7518626004

03 November 2020

Reviewer 3 Report

This article is written well, however, this need to revised a few points.  

1) What is the hypothesis of this study?

2) Please describe the methodology separately.

3) Is there a limit to this study? Please indicate the limits of your research.

Author Response

We are thankful to the reviewers who have kindly commented on our manuscript with the editorial number antibiotics-978192.

After a thorough and careful consideration of the reviewers’ comments, we have presented our itemised responses in a tabular form for your kind perusal.

Reviewers’ comments Our response Changes in revised manuscript

Reviewer 3

Point 1

Is there a limit to this study? Please indicate the limits of your research.

We have noted that both reviewers 2 & 3 have made a similar comment regarding the need for a more explicit declaration of the limitation(s) of our study. We have added a number of subsections with informative headings that summarise key points in the discussion.

To address the limitations, we have added the below three main plausible limitations:

3.3. Limitations of the study

3.3.1 Generalizability and transferability of the tool

This survey tool was developed with a focus on the current community pharmacy practices in Thailand, where community pharmacists can legally prescribe/ dispense antibiotics. Prior to using this survey tool elsewhere, cross-cultural adaptation and validation should be undertaken. We would highly recommend the use of this tool to the researchers in their future work, where it should be utilized in a representative sample of community pharmacists as well as hospital pharmacists.

3.3.2 Possible measurement bias as a result of applying AI theory:

It should also be noted that, AI, by definition, focuses on the strengths of the current antibiotic prescribing/dispensing practices and does not focus on the problems/issues. Hence this tool could be less sensitive in identifying and measuring the current shortcomings of the antibiotic prescribing/dispensing practices of the community pharmacists. Nevertheless, it is important to re-emphasize the usefulness of this tool as a good starting point bring about behavior change, an affordable and a sustainable intervention, to tackle antimicrobial resistance in developing countries where the resources are scarce.

3.3.3 Participant recruitment bias

In response to the current COVID-19 pandemic, we had to collect data via Facebook and Line, instead of using the traditional paper and pencil method of data collection. It has been shown that online data collection, specially via social media platforms could introduce recruitment bias as the use of such platforms have been associated with certain socio-demographic characteristics [41]. However, after analysing the participants, characteristics (See table 3 - 7) it seemed that the participants represented Thai community pharmacists well [42].

Refer to the lines 234 to 255 on pages 14 under the discussion section to view the track changes.

Reviewer 3

Point 2

What is the hypothesis of this study?

We appreciate the reviewer’s comment and the hypothesis of this study is Appreciate Inquiry could be used to create tools to quantitatively measure community pharmacists’ attitudes towards antibiotic smart use in Thailand.

Refer to lines 272 - 283 on page 15 in the methods section.

Reviewer 3

Point 3

Please describe the methodology separately.

We thank the reviewer for pointing this out. We have made the necessary changes by separating and rearranging the sub-headings in the methodology section. We have added more detail about pilot test under the sub-heading 4.4 “Pilot test” in line 328 – 333 on page 17 and data analysis under the sub-heading 4.6 “Data Analysis” in line 344 – 350 on page 18. Refer to page 15 - 18 of the manuscript.

Regarding more minor matters, we have now added some sentences and have changed order of the authors list, our spelling and phrasing patterns. The changes are highlighted using track changes option as well as being highlighted in yellow. We have provided the line number along with page number for each response, where possible, to facilitate navigation through the revised manuscript.

Keywords

We have added “community pharmacist”. (line 32, page 1)

Introduction

The introduction could be much more clear. We have included sentences such as:

“The main concerns are the inadequacy of surveillance systems and non-prescription of antibiotic use [11].” (line 39 – 40, page 1)

“Challenges to combatting antimicrobial resistance are inadequate surveillance systems, lack of antimicrobial consumption data, a lack of governance support related to policy research and incomprehensive antibiotic stewardship [12].” (line 42 – 44, page 2)

“Around 13,906 community pharmacies are private and provide a convenient first point of contact between patients and the health care system [17]. Community pharmacists play a vital role in providing healthcare services for people in community due to the convenience of their location and short waiting times [18].” (line 56 – 59, page 2)

“There is a dearth of published literature pertaining to AI in the context of community pharmacy and antibiotic practice among healthcare professionals [24,26–28].“ (line 67 – 69, page 2)

“Although other studies have developed tools to measure self-medication with antibiotics among the general public” (line 79 – 80, page 2)

Discussion

To address the strength of the novel survey tool, a new paragraph has been added to sub-heading 3.1. “Survey tool design and development “ in line 217 -222 on page 13.

“To the best of our knowledge, AI theory has not previously been applied in community pharmacy research. This is the first research to literature the AI approach in the community pharmacy setting [24,26–28]. It is strength-based research that offers people in the community or organization to share their knowledge and build new opportunities in their community or their organization. Especially, community pharmacists who are professional pharmacy services in community pharmacy, must be willing about caring professionals and enhancing quality of their services.”

We hope the revised manuscript will better suit the Antibiotics but are happy to consider further revisions, and we thank you for your continued interest in our research.

Sincerely,

Rojjares Netthong (On behalf of the Research Team)

School of Pharmacy

University of Lincoln,

Joseph Banks Laboratories

Beevor St, Lincoln LN6 7DL, United Kingdom

Mobile Tel: +44 7518626004

03 November 2020

Round 2

Reviewer 1 Report

Dear Authors,

I have read the revised version of the original paper ID: antibiotics-978192-peer-review-v2 entitled “Antimicrobial Resistance, Pharmacists and Appreciative Inquiry: Development of a Novel Measurement Tool”. I think, that the Authors have made extensive revisions of this article according to the suggestions of all reviewers. This manuscript is worth publishing in “Antibiotics”.

With highest regards,